# Hepatitis B virus hijacks TSG101 to facilitate egress via multiple vesicle bodies

Yingcheng Zheng[1☉], Mengfei Wang[1☉], Sitong Li[1☉], Yanan Bu[1], Zaichao Xu[1], Guoguo Zhu[2], Chuanjian Wu[1], Kaitao Zhao[1], Aixin Li[1], Quan Chen[1], Jingjing Wang[1], Rong Hua[1], Yan Teng[1], Li Zhao[1], Xiaoming Cheng[1,3,4], Yuchen Xia [1] *

1 State Key Laboratory of Virology and Hubei Province Key Laboratory of Allergy and Immunology, Hubei Jiangxia Laboratory, Institute of Medical Virology, TaiKang Center for Life and Medical Sciences, TaiKang Medical School, Wuhan University, Wuhan, China, 2 Department of Emergency, General Hospital of Central Theater Command of People's Liberation Army of China, Wuhan, China, 3 Wuhan University Center for Pathology and Molecular Diagnostics, Zhongnan Hospital of Wuhan University, Wuhan, China, 4 Hubei Clinical Center and Key Laboratory of Intestinal and Colorectal Diseases, Wuhan, China

☉ These authors contributed equally to this work.
* yuchenxia@whu.edu.cn

**Data Availability Statement:** All relevant data are within the manuscript and its Supporting information files.

**Funding:** This work was supported by the National Natural Science Foundation of China (project no.

## Abstract

Hepatitis B virus (HBV) chronically infects 296 million individuals and there is no cure. As an important step of viral life cycle, the mechanisms of HBV egress remain poorly elucidated. With proteomic approach to identify capsid protein (HBc) associated host factors and siRNA screen, we uncovered tumor susceptibility gene 101 (TSG101). Knockdown of TSG101 in HBV-producing cells, HBV-infected cells and HBV transgenic mice suppressed HBV release. Co-immunoprecipitation and site mutagenesis revealed that VFND motif in TSG101 and Lys-96 ubiquitination in HBc were essential for TSG101-HBc interaction. *In vitro* ubiquitination experiment demonstrated that UbcH6 and NEDD4 were potential E2 ubiquitin-conjugating enzyme and E3 ligase that catalyzed HBc ubiquitination, respectively. PPAY motif in HBc and Cys-867 in NEDD4 were required for HBc ubiquitination, TSG101-HBc interaction and HBV egress. Transmission electron microscopy confirmed that TSG101 or NEDD4 knockdown reduces HBV particles count in multivesicular bodies (MVBs). Our work indicates that TSG101 recognition for NEDD4 ubiquitylated HBc is critical for MVBs mediated HBV egress.

## Author summary

HBV virion assembly is initiated with nucleocapsid transportation to the surface of the MVBs, then buds into MVB through the endosomal sorting complex required for transport (ESCRT) complexes on contact with the HBV envelope proteins via endosomal sorting complex. However, it is still unclear by which and how host factor(s) recognizes HBV virions and sorts them into MVBs to initiate the egress pathway. This study shows that TSG101 recognizes ubiquitinated HBc via its VFND sequence and gives access for HBV to the MVBs egress route. Furthermore, we find that the Lys-96 residue and PPAY motif in HBc are essential for bridging HBc and TSG101. We also demonstrated that the

81971936 to YX), Hubei Province's Outstanding
Medical Academic Leader Program (to YX),
Foundation for Innovative Research Groups of the
Natural Science Foundation of Hubei (project no.
2020CFA015 to YX), the Fundamental Research
Funds for the Central Universities (project no.
2042022kf1215 to XC and 2042021gf0013 to YX)
and Basic and Clinical Medical Research Joint Fund
of Zhongnan Hospital, Wuhan University (to YX).
The funders had no role in study design, data
collection and analysis, decision to publish, or
preparation of the manuscript.

**Competing interests:** The authors have declared
that no competing interests exist.

endosomal E3 ubiquitin ligase NEDD4 catalyze HBc ubiquitination and the PPAY motif
in HBc is required for NEDD4 recruitment. In sum, our findings provide insight into the
molecular mechanism of how HBV virions were sorted into MVBs to egress.

## Introduction

Hepatitis B virus (HBV) chronically infects 296 million individuals and currently there is no
cure [1]. Those individuals are at high risk of developing cirrhosis and hepatocellular carci-
noma [2]. HBV is an enveloped virus encodes three viral surface proteins, namely large (L),
middle (M) and small hepatitis B virus surface protein (S). They are integral membrane pro-
teins which surround the viral nucleocapsid assembled by the capsid protein (HBc). The nucle-
ocapsid harbors a 3.2 kb partially double-stranded relaxed circular DNA (rcDNA) genome. By
attaching to heparan sulfate proteoglycans [3,4] and interacting with sodium taurocholate co-
transporting polypeptide (NTCP) [5], HBV enters into the hepatocyte and HBV genome is
delivered into the nucleus where it is repaired and chromatinized to form an episomal cova-
lently closed circular DNA (cccDNA) [6]. The cccDNA encodes 5 overlapping viral RNAs
which lead to expression of viral structure and non-structure proteins [7]. Particularly, prege-
nomic RNA (pgRNA) is encapsidated together with the viral polymerase (Pol). Inside the cap-
sid, HBV rcDNA genome is generated through reverse transcription. At late stage of HBV
infection, the viral particles are assembled and leave the cell. The intact viral particles named
Dane particles are released via late endosomal compartments multivesicular bodies (MVBs)
[8]. The extracellular export is essential for infection transmission. HBV virion assembly is ini-
tiated with nucleocapsid transportation to the surface of the MVBs, through NEDD4, α-taxilin,
and then buds into MVB through endosomal sorting complex required for transport (ESCRT)
complexes on contact with the HBV envelope proteins via endosomal sorting complex [9–11].
MVB and/or MVB-derived vesicles then fuse with the plasma membrane to release HBV
virion [12]. However, it is still unclear which and how host factors recognize HBV capsids and
facilitate the morphogenesis and egress of HBV virions from hepatocytes.

To dissect the molecular mechanisms in HBV egress, we took advantage of immunoprecipi-
tation-mass spectrometry assay and identified TSG101 as a HBc binding partner. Mechanistic
study revealed that after encapsidation, NEDD4 was recruited to HBc and catalyzed the ubi-
quitination of HBc. By TSG101 recognition, HBV capsids were sorted into MVBs and for
extracellular export.

## Results

### Identification of TSG101 as regulator of HBV egress and HBV antigens secretion

To investigate host factor(s) involved in HBV egress, we first aimed to identify proteins inter-
acting with HBc. We transfected N-terminal tagged Myc-HBc plasmid into Huh7 cells and
performed immunoprecipitation-mass spectrometry (Fig 1A). Native agarose gel immunoblot
analysis revealed that Myc-HBc was capable of capsid formation (S1A Fig). Mass spectrometry
identified 92 proteins that may interact with capsid or HBc (S1B Fig). GO analysis of biological
process and cellular component for the identified HBc-associated factors showed that they
were associated with "protein transport", "viral process" and located in "cytoplasm" and
"extracellular exosome" (S1C–S1D Fig). As an overlap between virus egress and exosome bio-
genesis were reported [13], this data suggested that HBc binder might regulate virus egress. So,

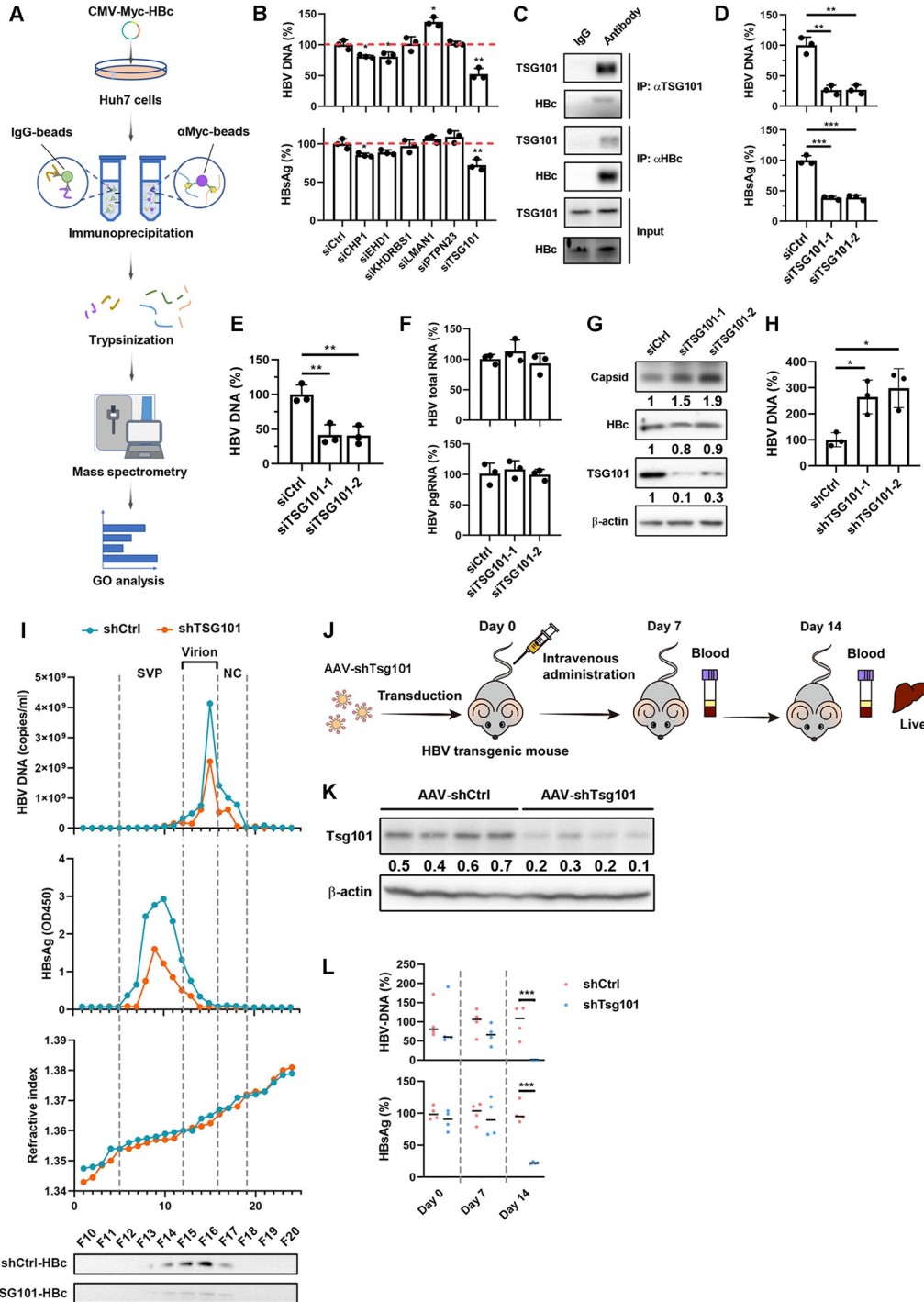

**Fig 1. TSG101 is a potential host factor involved in HBV egress.** (A) Schematic view of the immunoprecipitation-mass spectrometry assay procedures. Myc-HBc expressing plasmid was transfected into Huh7 cells. Myc IP was conducted on the cell lysates. IgG IP was set as negative control. The immunoprecipitated proteins were trypsinized to peptides and were separated by HPLC before mass spectrometry. (B) HepAD38 cells were transfected with CHP1, EHD1, KHDRBS1, LMAN1, PTPN23, or TSG101 targeted siRNA respectively and maintained with DMEM containing 2% DMSO for 2 days. Levels of HBV DNA and HBsAg in cell culture supernatant were determined by qPCR and ELISA respectively (% of siCtrl). (C) HepG2-NTCP cells were pretreated with 2.5% DMSO for 2 days following HBV infection at an MOI of 200 and maintained with 2.5% DMSO for 7 days. The cells were lysed with IP buffer. HBc IP and TSG101 IP experiments were performed on the cell lysates. IgG IP was set as negative control. (D-F) HepG2-NTCP cells transfected with TSG101-targeted siRNA were pretreated with 2.5% DMSO for 2 days following HBV infection at an MOI of 200 and maintained with 2.5% DMSO for 5

days. (D) Levels of HBV DNA, HBsAg in cell culture supernatant were determined by qPCR and ELISA respectively (% of siCtrl). (E) Levels of HBV DNA in HBs antibody immunoprecipitation of culture medium were determined by qPCR. (F) Levels of intracellular HBV total RNA and pgRNA were determined by qPCR (% of siCtrl). Values show the mean ± SD, ***$p < 0.001$. (G) Levels of TSG101 and HBc were determined by WB. Levels of HBV capsids in liver were detected by agarose gel electrophoresis-immunoblot. (H) Lysates from HepAD38 cells with TSG10 knockdown. Capsids were immunoprecipitated. HBV DNA levels were determined by qPCR after capsid disruption. (I) The culture supernatant of HepAD38 cells with or without TSG101 knockdown was concentrated with Amicon Ultra-15 centrifugal filter unit and subjected to CsCl density gradient centrifugation. Levels of HBV DNA HBsAg, and HBc in each fraction were determined by qPCR, ELISA, and l WB respectively. (J) Timeline of AAV transduction of the HBV transgenic mice and specimen collection. HBV transgenic mice were administrated with recombinant AAV expressing control shRNA or Tsg101-targeted shRNA ($1 \times 10^{12}$ viral genome) via tail intravenous injection. Orbital venous blood was collected at 7 days and 14 days after AAV administration. The mice were sacrificed 14 days after AAV administration and livers were harvest. (K) Knockdown efficiency of AAV-shTsg101 in liver were determined by WB. (L) Levels of serum HBV DNA and HBsAg were determined by qPCR and ELISA respectively (% of shCtrl). Values show the mean ± SEM, ***$p < 0.001$.

we selected the common factors from "protein transport", "viral process", and "extracellular exosome" categories for further investigation. To assess their roles in HBV life cycle, we conducted siRNA knockdown of each gene in HepAD38 cells, an HBV expressing cell line [14], and evaluated viral antigens and particles secretion. Among all 6 selected proteins, TSG101 knockdown led to a significant decrease of HBsAg as well as HBV DNA (Fig 1B), which mainly represents progeny virions in cell culture supernatant. The knockdown efficiency of individual siRNA was determined by qPCR (S1E Fig). These data suggested that TSG101 may participate in HBV egress.

## TSG101 interacts with HBc and mediates HBV egress

To confirm the interaction between HBc and TSG101, we performed co-immunoprecipitation (Co-IP) using HepAD38 cells and HBV infected HepG2-NTCP cells. In agreement with the result from Huh7 cells with overexpression of Myc-tagged HBc, wild type HBc co-immunoprecipitated with TSG101 (S2A and 1C Figs).

Next, HepG2-NTCP cells were first transfected with siRNA against TSG101, and then infected with HBV. Consistent with the previous results from HepAD38, knockdown of TSG101 led to significant decrease of HBV DNA and HBsAg (Fig 1D). To evaluate the effects of TSG101 knockdown on virion secretion, we determined levels of HBV DNA after HBsAg antibody immunoprecipitation of culture media from HBV infected HepG2-NTCP cells. The data showed that TSG101 knockdown reduced HBV DNA levels of HBsAg antibody immunoprecipitated viral particles in culture medium (Fig 1E). Of note, intracellular HBV total RNA, pgRNA, and HBc remain unaffected (Fig 1F and 1G). Interestingly, TSG101 knockdown resulted in elevated intracellular level of capsids and encapsided HBV DNA (Fig 1G and 1H), suggesting retardation of capsids extracellular export. Similar results were observed in HepAD38 cells and HBV infected HepG2-NTCP cells with stable TSG101 knockdown by shRNA (S2B–S2I Fig). To further prove that the secretion of viral particle was regulated by TSG101, supernatant of HepAD38 cells with or without TSG101 knockdown was concentrated and subjected to CsCl density gradient centrifugation. Different markers from various fractions were evaluated. As shown in Fig 1I, the fractions represent HBV virion showed decreased HBV DNA, HBsAg and HBc levels. To investigate why the retarded capsid by TSG knockdown did not result in intracellular HBc elevation, cells were treated with proteasome inhibitor MG132 or autophagy inhibitor Bafilomycin A1. Interestingly, blockade of autophagy but not proteasome led to enhanced intracellular HBc signal even after TSG101 knockdown (S2J Fig). These data suggested that intracellular level of HBc is regulated by autophagy.

To further confirm the role of TSG101 *in vivo*, HBV transgenic mice were injected with adeno-associated virus (AAV) expressing Tsg101 shRNA (Fig 1J). Western blot (WB) verified

that Tsg101 was efficiently reduced (Fig 1K). As expected, AAV-shRNA targeting Tsg101 significantly suppressed HBsAg and HBV DNA in the serum of HBV transgenic mice (Fig 1L). Together, these data indicated that TSG101 could interact with HBc and facilitate HBV egress.

## TSG101 regulates the MVB-dependent HBV egress pathway

TSG101 is a component of the ESCRT-I complex and functions in vacuolar protein sorting [15]. It has been reported that HBV virions bud into MVB through ESCRT complexes [8,9,11]. However, how the viral particles are sorted into MVBs is poorly elucidated. Based on our data, we speculated that the interaction of TSG101 with HBc was critical for HBV sorting into MVBs. We next investigated the effects of HBV on the maturation of $TSG101^+CD63^+$ vesicles. By comparing HepAD38 cells with or without active HBV replication, we found that HBV production resulted in increased number of $TSG101^+CD63^+$ vesicles (S3A Fig). To further investigate the role of HBV in MVBs maturation, we analyzed the subcellular distribution of CD63 and HBc in HepAD38 cells. As shown in Fig 2A, in HBV expressing cells, HBc colocalized with the CD63-labeled MVBs, which was further confirmed by Pearson correlation coefficient method. Next, we examined the ultracellular structure of MVBs by transmission electron microscopy (TEM). MVBs containing HBV particles could be observed in HBV expressing HepAD38 cells (Fig 2B). MVBs in HepAD38 showed larger size when HBV replication was activated (Fig 2B) which suggested that HBV improved the maturation of MVBs.

To confirm that MVBs sorting is a key step in HBV egress, we treated HBV infected HepG2-NTCP cells with U18666A, an inhibitor of MVB biogenesis [16]. The treatment of U18666A at 5 μg/ml significantly reduced secreted levels of HBV markers, especially HBV DNA (S3B Fig). While HBV total RNA and pgRNA remained stable (S3C Fig), the levels of intracellular capsids were elevated (S3D Fig). Vitality assay proved the effects observed were not caused by altered cell viability (S3E Fig).

To explore the role of TSG101 in MVBs sorting of HBV particles, we knocked down TSG101 by shRNA and evaluated the localization of HBV and MVBs by immunostaining. As shown in Fig 2C, the colocalization of HBc and CD63 was abolished after TSG101 knockdown. TEM further revealed that the knockdown of TSG101 led to reduced HBV particles count in MVBs and decreased MVBs size (Fig 2D). To further confirm the morphological identification of MVBs, we performed focused ion beam scanning electron microscope (FIB-SEM). The data showed that TSG101 knockdown reduced the size of MVBs in HepAD38 cells. (Fig 2E, Video 1 and 2 in https://figshare.com/s/d04bc90eb146340e231e), suggesting that TSG101 regulated MVBs maturation in HBV expressing cells. HBV particles were verified by immuno-electron microscopy (Fig 2F).

Together, these data suggested that TSG101 was a crucial factor mediating HBV egress through MVBs.

## Ubiquitin recognition domain is essential for TSG101 mediated HBV egress

TSG101 functions in the vacuolar protein sorting pathway, where its ubiquitin-E2-like variant (UEV) domain binds to ubiquitylated proteins as they are sorted into MVBs [17]. Based on our observations that HBc interacted with TSG101, we speculated that ubiquitin recognition of TSG101 was necessary for this interaction. It has been reported that the residues Val-43 [18], Asn-45 [18–20], Asp-46 [18] of UEV are essential for TSG101-ubiquitin binding. We thus mutated all these amino acids to Ala and investigated the interaction between TSG101 and HBc. Co-IP assay showed that the mutant $TSG101_{VFND43AAAA}$ did not interact with HBc (Fig 3A), suggesting that TSG101-ubiquitin binding was essential for TSG101-HBc interaction.

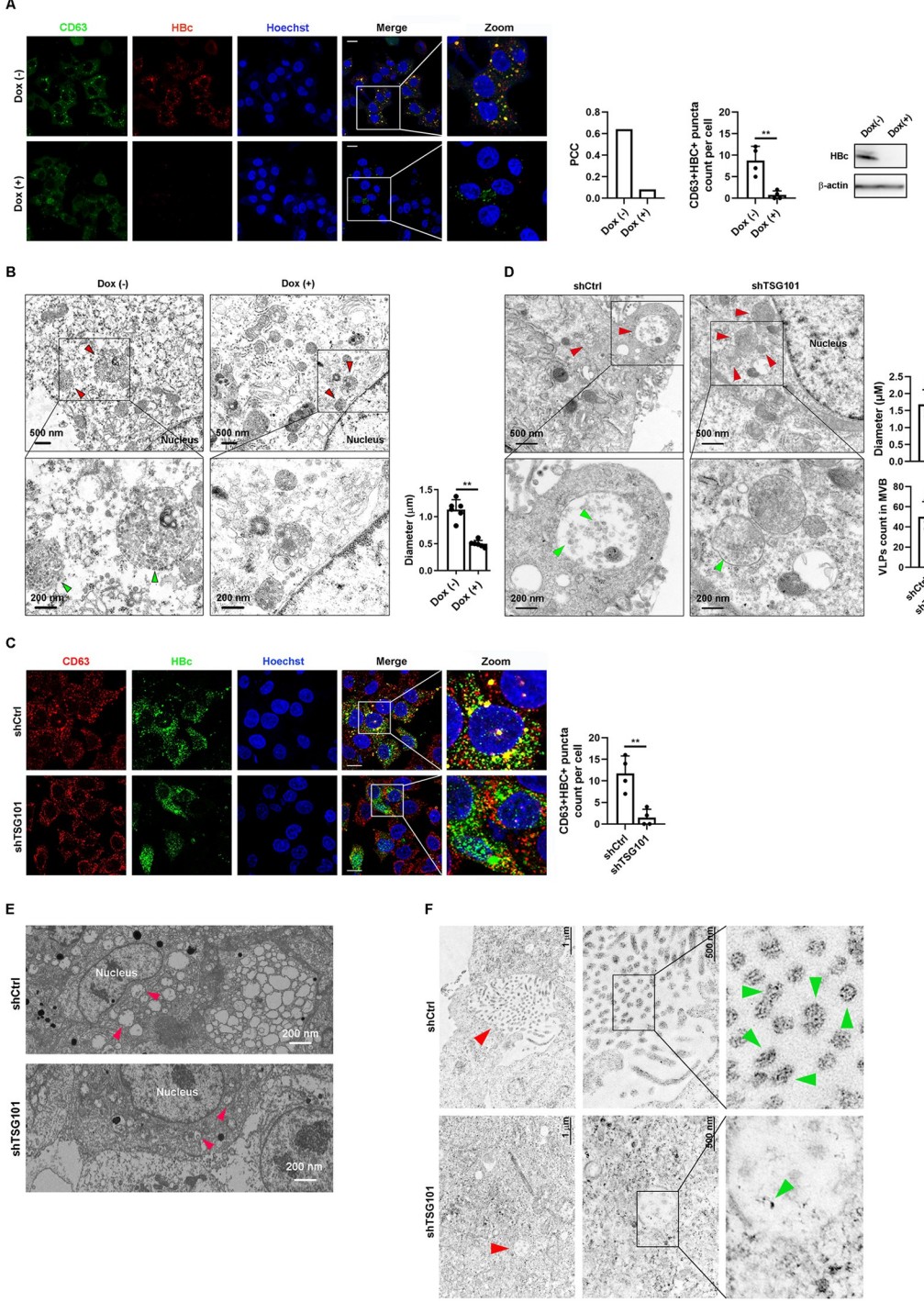

**Fig 2. TSG101 is required for enrichment of HBV in MVB.** (A and B) HepAD38 cells were treated with or without doxycycline 1 µg/ml for 10 days. (A) Subcellular distribution of HBc and CD63 were detected by IF with confocal microscope. Scale bar = 10 µm. Colocalization ratio was determined in ImageJ by the Pearson correlation coefficient method (PCC). Counts of CD63[+]HBc[+] puncta. Levels of HBc were determined by WB. (B) Transmission electron microscopy (TEM) was used for cellular ultrastructural examination of HBV like particles (green arrow) and MVBs (red arrow). Quantification of MVB diameters and counts of virion-like particle (VLP) in MVB. (C and D) HepAD38 with or without stable knockdown of TSG101. (C) Subcellular distribution of CD63 and HBc detected by IF with confocal microscope. Scale bar = 10 µm. Counts of CD63[+]HBc[+] puncta. (D) TEM was used for cellular ultrastructural examination of HBV particles (green arrow) and MVBs (red arrow) in HepAD38 with or without stable knockdown of TSG101. Quantification of MVB diameters and

counts of VLP in MVB. Values show the mean ± SEM, *p < 0.05, **p < 0.01. (E) Representative images of FIB-SEM assay for MVBs (red arrow) in HepAD38 cells with TSG101 knockdown. (F) HBc antibody conjugated to 4 nm gold particle. Representative images of immunoelectron microscopy for HBc-labeled particles (green arrow) in HepAD38 with or without stable knockdown of TSG101. Red arrow indicates MVBs.

We then reconstituted wild type or mutant TSG101 in TSG101 stable knockdown cells and determined different HBV markers. The expression of different forms of TSG101 were tested by western blot (Fig 3B). According to the levels of HBV DNA, we found that only complementation of wild type TSG101, but not VFND43AAAA mutant, could completely rescue the levels of HBV DNA (Fig 3C). However, overexpression of TSG101 had no additional effect suggesting that the baseline expression of TSG101 was sufficient to mediate HBV egress (Fig 3C). As expected, intracellular HBV RNAs were not affected by either wild type or mutant TSG101 (Fig 3D).

Collectively, these results suggested that ubiquitin recognition domain of TSG101 was essential for the interaction with HBc and HBV egress.

## HBc K96 ubiquitination is critical for TSG101 binding and HBV egress

As ubiquitin recognition was required for TSG101 mediated HBV egress and HBc interaction, we reasoned that ubiquitination of HBc was obligatory. We analyzed HBc sequences from different HBV genotypes and found two conserved canonical ubiquitination sites, K7 and K96 (S4A Fig). To further dissect the mechanism involved in HBV egress, we created K7R, K96R, and K7/96R HBc mutants and investigated their binding abilities with TSG101. As shown in Fig 4A, only K7R mutant, but not K96R or K7/96R mutant, behaved like wild type HBc and interacted with TSG101. These results suggested that the K96 ubiquitination was indispensable for HBc-TSG101 interaction. Next, we investigated the effect of HBc ubiquitination on HBV egress. An AAV-HBV1.2ΔHBc plasmid harboring a 1.2-fold HBV genome deficient in HBc

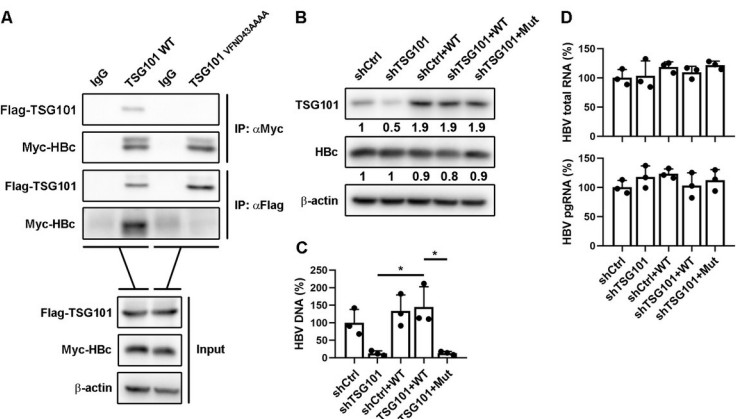

**Fig 3. VFND motif is critical for TSG101 to regulate HBV egress.** (A) Flag-TSG101 or Flag-TSG101$_{VFND43AAAA}$ expressing plasmid was co-transfected with Myc-HBc expressing plasmid into Huh7 cells. Myc IP and Flag IP experiments were conducted on the cell lysates. IgG IP was set as negative control. (B-D) Stable TSG101 knockdown HepG2-NTCP cells (prepared with lentivirus expressing TSG101 mRNA 3'UTR-targeted shRNA) were infected with lentivirus expressing TSG101 or TSG101$_{VFND43AAAA}$. The cells were pretreated with 2.5% DMSO for 2 days following HBV infection at an MOI of 200 and maintained with 2.5% DMSO for 7 days. (B) Levels of TSG101 and HBc were determined by WB. (C) Levels of HBV DNA in cell culture supernatant were determined by qPCR (% of shCtrl). (D) Levels of intracellular HBV total RNA and pgRNA were determined by qPCR (% of shCtrl). Values show the mean ± SD. *p < 0.05.

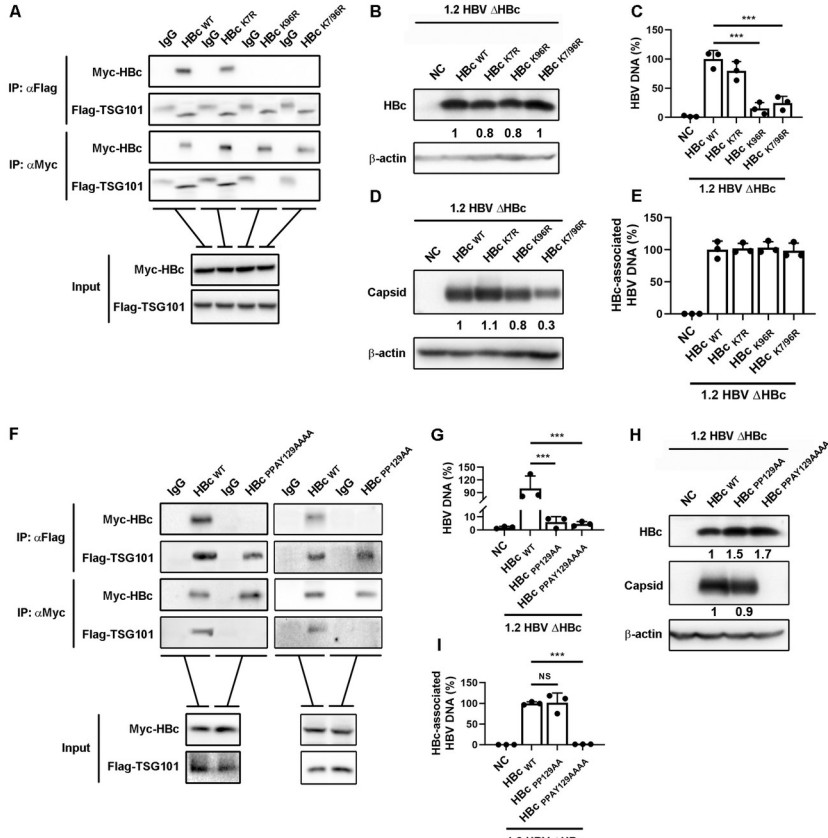

**Fig 4. K96 and PPAY motif in HBc are key for TSG101 recognition and HBV egress.** (A) Flag-TSG101 expressing plasmid was co-transfected with Myc-HBc, Myc-HBc$_{K7R}$, Myc-HBc$_{K96R}$, or Myc-HBc$_{K7/96R}$ expressing plasmid into Huh7 cell. Myc IP and Flag IP experiments were conducted on the cell lysates. IgG IP was set as negative control. (B-E) HBcHBc$_{K7R}$, HBc$_{K96R}$, or HBc$_{K7/96R}$ expressing plasmid was co-transfected with the plasmid pAAV-1.2HBVΔHBc into Huh7 cells. The cells and culture supernatants were harvest at 72 hours post transfection. (B) Levels of HBc were determined by WB. (C) Copies of HBV DNA in cell culture supernatant were determined by qPCR (% of HBc$_{WT}$). (D) Intracellular HBV capsids were detected by agarose gel electrophoresis-immunoblot. (E) Levels of HBc-associated HBV DNA were determined by qPCR. (F) Myc-HBc, Myc-HBc$_{PPAY129AAAA}$, or Myc-HBc$_{PP129AA}$ expressing plasmid was co-transfected with Flag-TSG101 expressing plasmid into Huh7 cells. Myc IP and Flag IP experiments were conducted on the cell lysates. (F and G) HBc, HBc$_{PP129AA}$, or HBc$_{PPAY129AAAA}$ expressing plasmid was co-transfected with the plasmid pAAV-1.2HBVΔHBc into Huh7 cells. The cells and culture supernatants were harvest at 72 hours post transfection. (G) Levels of HBV DNA in cell culture supernatant were determined by qPCR (% of HBc$_{WT}$). (H) Levels of HBc were determined by WB. Intracellular HBV capsids were detected by agarose gel electrophoresis-immunoblot. (I) Levels of HBc-associated HBV DNA were determined by qPCR. Values shows the mean ± SD, ***$p < 0.001$.

expression was co-transfected with plasmid expressing wild type HBc or HBc with K7R, K96R, or K7/96R mutation into Huh7 cells. The expression of HBc was validated by western blot (Fig 4B). We then accessed the secreted HBV DNA to determine the effect of HBc ubiquitination on HBV egress. Both K96R and K7/96R mutants, but not K7R mutant, led to a remarkable decrease of extracellular HBV DNA (Fig 4C). To exclude the possibility that the HBc mutations affected capsid formation thus resulted in less virion secretion, we examined intracellular HBV capsids by native gel electrophoresis-immunoblot assay. While only K7/96R mutation slightly affected capsid formation (Fig 4D), there was no significant difference in HBc-associated HBV DNA among the mutants (Fig 4E), further proved the role of HBc K96 ubiquitination in HBV egress. Together, these results indicated that HBc K96 ubiquitination was critical for TSG101 binding and HBV egress.

## The PPxY motif in HBc is crucial for HBV egress

It has been reported that Pro-Pro-x (any amino acid)-Tyr (PPxY) motif in the substrate protein is necessary for TSG101 recognition and binding via ubiquitin bridge [21]. We analyzed all HBV proteins and found that HBc contained a conserved PPxY motif (S4B Fig). We then created two HBc constructs with PPAY129AAAA or PP129AA mutation, respectively, and evaluated their interaction with TSG101. Not surprisingly, both HBc mutation constructs failed to interact with TSG101 (Fig 4F). Next, we investigated if the PPxY motif in HBc is essential for HBV egress. AAV-HBV1.2 ΔHBc plasmid was co-transfected with plasmid expressing wild type or mutant HBc. As demonstrated by HBV DNA presented in cell culture supernatant, only complementation with wide type HBc, but not PPAY129AAAA or PP129AA mutants, could rescue HBV egress (Fig 4G). We next explored if the PPxY mutations we created affected capsid formation thus resulted in impaired virion production. As shown in Fig 4H, only PPAY129AAAA, but not PP129AA mutant, was incapable of capsid formation. Along the same line, while HBc PPAY129AAAA failed to produce any HBc-associated HBV DNA, the PP129AA mutation behaved similar to wild type in intracellular HBV core DNA production (Fig 4I). The data suggested HBc Tyr-132 is critical for capsid assembly. Collectively, our results demonstrated that the PPxY motif in HBc was essential for HBc-TSG101 interaction and HBV egress.

## The E3 ligase NEDD4 is critical for HBV egress

Next question raised was how HBc was ubiquitylated. Protein ubiquitination involves several cellular enzymes including ubiquitin activating enzyme (E1), ubiquitin conjugating enzyme (E2) and ubiquitin ligase (E3), among which E3 ligase is responsible for substrate recognition [22]. It is known that several retroviral proteins recruit NEDD4 E3 ligase with their PPxY motif to carry out ubiquitination [23,24]. Therefore, we speculated that HBc ubiquitination was catalyzed by NEDD4 and subsequently recognized by TSG101 to mediate HBV virions extracellular export. There are two isoforms of human NEDD4, NEDD4 and NEDD4L [25]. To clarify which specific NEDD4 catalyzed HBc, we knocked down NEDD4 and NEDD4L respectively to investigate the effects on HBV life cycle. We first used siRNA to knockdown NEDD4 in HBV infected HepG2-NTCP cells. The results showed that NEDD4 knockdown significantly decreased HBV DNA and HBsAg in cell culture supernatants (Fig 5A) without affecting intracellular viral RNAs (Fig 5B). In alignment with the effects of TSG101 knockdown, the NEDD4 knockdown led to increased intracellular capsids and encapsidated HBV DNA levels (Fig 5C and 5D). Similar results were observed in HepAD38 cells and HBV infected HepG2-NTCP cells with NEDD4 stably knockdown by shRNA (S5A–S5F Fig). On the contrary, NEDD4L knockdown did not affect any HBV replication markers in HepAD38 cells and HBV infected HepG2-NTCP cells (S5G–S5L Fig). These data pointed out that it was NEDD4, but not NEDD4L, regulated HBV extracellular export.

The HECT domain is known to play essential role in NEDD4 mediated E3 ubiquitin ligation [26]. To further confirm NEDD4 functions as a E3 ligase to regulate HBV replication, we complemented NEDD4 stable-knockdown HepG2-NTCP cells with wild type or catalytic activity mutated NEDD4. The expression of NEDD4 was determined by western blot (Fig 5E). After HBV infection, the complementation of wild type NEDD4, but not the NEDD4 with C867S mutation, completely rescued the HBV DNA levels in culture supernatant (Fig 5F). Meanwhile, intracellular HBV total RNA and pgRNA remained unchanged (Fig 5G). TEM showed that the NEDD4 knockdown led to decreased number of viral particles in MVBs and reduced MVBs sizes (Fig 5H). These results suggested that the catalytic activity of NEDD4 was required for HBV egress.

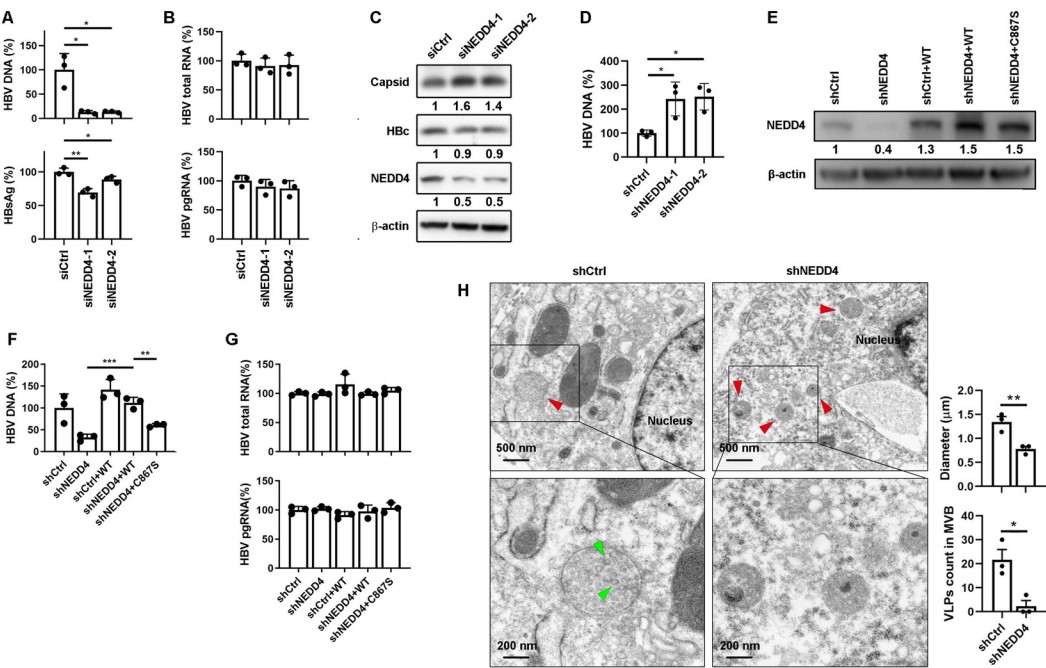

**Fig 5. NEDD4 is critical for HBV egress.** (A-C) HepG2-NTCP cells transfected with NEDD4 targeted siRNA were pretreated with 2.5% DMSO for 2 days following HBV infection at an MOI of 200 and maintained with 2.5% DMSO for 5 days. (A) Levels of HBV DNA and HBsAg in cell culture supernatant were determined by qPCR and ELISA respectively (% of siCtrl). (B) Levels of intracellular HBV total RNA and pgRNA were determined by qPCR (% of siCtrl). Values show the mean ± SD, *p < 0.05, **p < 0.01. (C) Levels of NEDD4 and HBc were determined by WB. Intracellular HBV capsids were detected by agarose gel electrophoresis-immunoblot. (D) Lysates from HepAD38 cells with NEDD4 knockdown. Capsids were immunoprecipitated. HBV DNA levels were determined by qPCR after capsid disruption. (E-G) Stable NEDD4 knockdown HepG2-NTCP cells (prepared with lentivirus expressing NEDD4 mRNA 3'UTR-targeted shRNA) were infected with lentivirus expressing NEDD4 or NEDD4$_{C867S}$. The cells were pretreated with 2.5% DMSO for 2 days following HBV infection at an MOI of 200 and maintained with 2.5% DMSO for 7 days. (E) Levels of NEDD4 were determined by WB. (F) Levels of HBV DNA in cell culture supernatant were determined by qPCR (% of HBc$_{WT}$). (G) Levels of intracellular HBV total RNA and pgRNA were determined by qPCR (% of HBc$_{WT}$). Values show the mean ± SD, **p < 0.01, ***p < 0.001. (H) Cellular ultrastructural examination of HBV like particles (green arrow) and MVBs (red arrow) in HepAD38 with or without stable knockdown of NEDD4 was detected by TEM. Quantification of MVB diameters and counts of VLP in MVB. Values show the mean ± SEM, *p < 0.05, **p < 0.01.

## NEDD4 mediates the ubiquitination of HBc

To verify the role of NEDD4 in HBc ubiquitination, we first accessed the interaction of NEDD4 and HBc in HepAD38 cells by Co-IP (S5M Fig). In HBc and NEDD4 Co-transfected Huh7 cells, Co-IP assay revealed that PP129AA failed to interact with NEDD4 (Fig 6A), suggesting that NEDD4 interaction with HBc was PPAY motif-dependent. Further, we immunoprecipitated HBc in HepAD38 cells with or without NEDD4 knockdown and detected the level of ubiquitination. As expected, the NEDD4 knockdown restrained the ubiquitination levels of HBc (Fig 6B). To confirm if the polyubiquitylation of HBc depends on the 129PPAY132 motif, we immunoprecipitated wild type HBc and HBcPP129AA from transfected Huh7 with NEDD4 knockdown, and detected the levels of ubiquitination by WB. The results showed that 129PPAY132 mutant demonstrated strongly suppressed ubiquitination level in the presence of NEDD4, and knockdown of NEDD4 only reduced the levels of wild type HBc ubiquitination but not PP129AA mutant (Fig 6C), suggesting that PPAY motif was critical for HBc ubiquitination. To validate the role of NEDD4 in HBc ubiquitination, an *in vitro* ubiquitination experiment was performed with purified HBc and NEDD4. As shown in Fig 6D, after reacted with

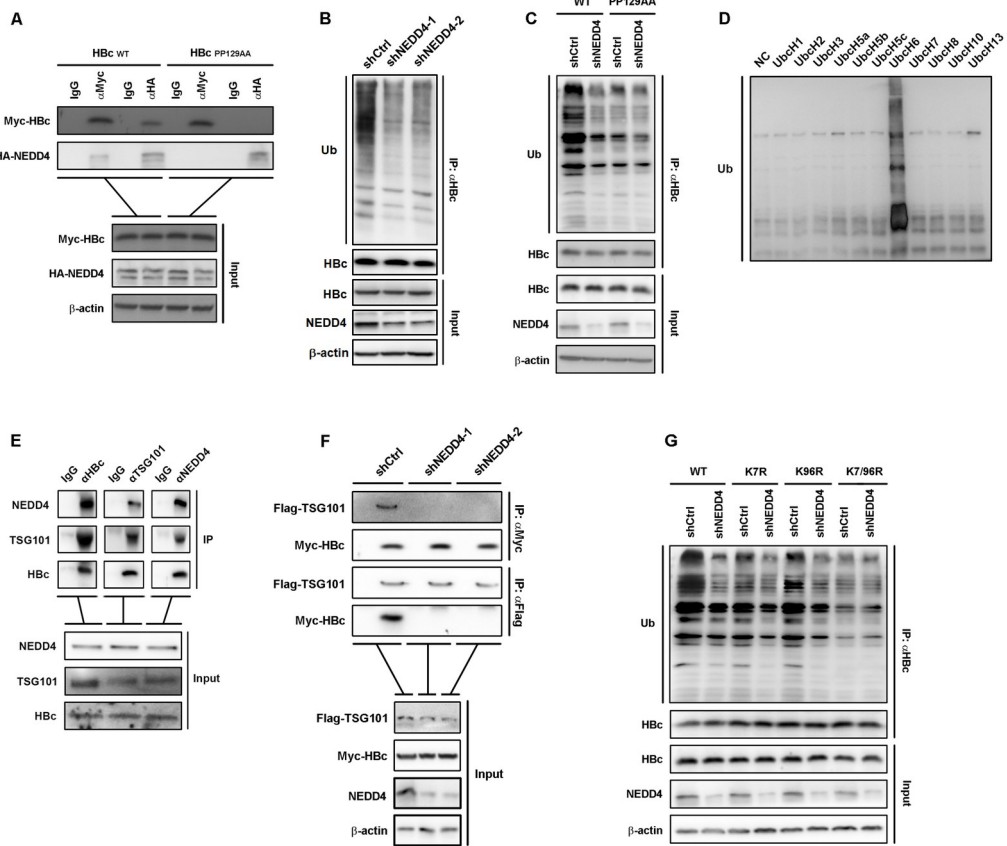

**Fig 6. NEDD4 mediates the ubiquitination of HBc.** (A) Myc-HBc$_{WT}$ or Myc-HBc$_{PP129AA}$ was co-transfected with HA-NEDD4 into Huh7 cells. The interaction between HBc and NEDD4 was detected by Co-IP assay. (B) Ubiquitin blots following HBc IP on the lysates of HepAD38 cells with or without stable NEDD4 knockdown. (C) HBc$_{PP129AA}$ and HA-ubiquitin were co-transfected into Huh7 cells with NEDD4 knockdown. Ubiquitination levels of immunoprecipitated HBc were determined by WB. (D) WB of *in vitro* ubiquitination of purified HBc by a panel of different E2 ubiquitin conjugating enzymes with *in vitro* purified NEDD4. (E) *In vitro* purified HBc was ubiquitinated by the E2 ubiquitin conjugating enzyme UbcH6 with *in vitro* purified NEDD4. Co-IP assay for interaction between *in vitro* purified TSG101 UEV domain, NEDD4, and the ubiquitinated HBc. (F) Flag-TSG101 expressing plasmid was co-transfected with Myc-HBc expressing plasmid into Huh7 cells with or without stable NEDD4 knockdown. Myc IP and Flag IP experiments were performed on the cell lysates. IgG IP was set as negative control. (G) HBc$_{K7R}$, HBc$_{K96R}$, or HBc$_{K7/96R}$ was co-transfected with HA-ubiquitin into Huh7 cells with NEDD4 knockdown. Ubiquitination levels of immunoprecipitated HBc were determined by WB.

E2 ubiquitin-conjugating enzyme UbcH6 and NEDD4, HBc was successfully ubiquitinated. These data indicated that UbcH6 and NEDD4 were potential E2 ubiquitin-conjugating enzyme and E3 ligase that catalyzed HBc ubiquitination, respectively. The interactions among UbcH6 treated HBc, NEDD4 and TSG101 were also evaluated by Co-IP *in vitro* (Fig 6E). To test if the NEDD4 mediated HBc ubiquitination was a prerequisite for TSG101-HBc interaction intracellularly, we co-transfected Flag-TSG101 and Myc-HBc into Huh7 cells with or without NEDD4 knockdown. As shown in Fig 6F, NEDD4 knockdown abolished the interaction between TSG101 and HBc. To further dissect which lysine residue was catalyzed by NEDD4, we investigated ubiquitination levels of HBc$_{K7R}$, HBc$_{K96R}$, and HBc$_{K7/96R}$. Interestingly, while both K7R and K96R mutants showed mitigated ubiquitination levels, K7/96R presented dramatic loss of ubiquitination signal (Fig 6G). NEDD4 knockdown further inhibited the ubiquitination levels of wild type HBc, HBc$_{K7R}$, and HBc$_{K96R}$ (Fig 6G), suggesting that

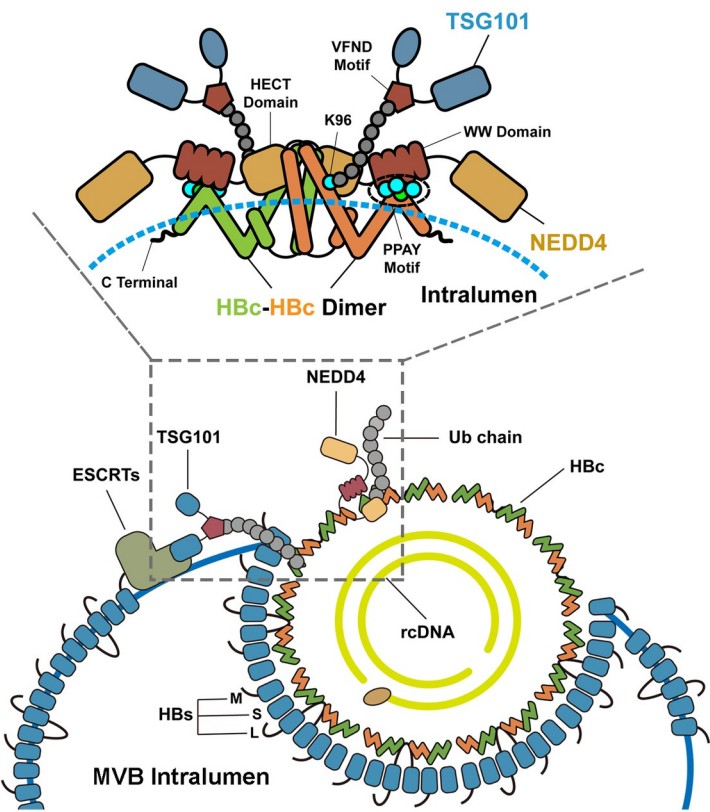

**Fig 7. Schematic summary.** TSG101 recognizes ubiquitylated HBc to give access for HBV to the MVB egress route. The ubiquitination of HBc is mediated by the endosomal E3 ubiquitin ligase NEDD4. Mechanically, NEDD4 is recruited by HBc via its PPAY motif and carry out the ubiquitination of HBc. TSG101 binds to ubiquitin linked HBc via its VFND sequence. The Lys-96 linked ubiquitin is essential for bridging TSG101 and HBc. The recognition of ubiquitylated HBc by TSG101 gives access for HBV to the MVBs egress pathway.

both K7 and K96 residue ubiquitination were catalyzed by NEDD4. Put together, our data suggested that E3 ligase NEDD4 mediated HBc ubiquitination was critical for TSG101-HBc interaction.

In sum, the initiation of HBV egress involves host factors TSG101 and NEDD4. As illustrated in Fig 7: the endosomal E3 ubiquitin ligase NEDD4 is recruited by HBc via its PPAY motif to carry out the ubiquitination of HBc at K96 site. The VFND motif of TSG101 then recognizes ubiquitylated HBc. As a core component of the ESCRTs complex, TSG101 grasps and delivers HBV capsid to MVB to facilitate virus assembly and egress.

## Discussion

It has been described that HBV particles are released through MVBs pathway [8], but how they are sorted into MVBs remains unclear. To decipher the molecular mechanisms, we conducted mass spectrometry to analyze host factors interact with HBc. Followed by signaling analysis and siRNA screen, we identified TSG101was highly likely to be involved. TSG101 recognized ubiquitylated HBc by interacting with ubiquitin at its VFND sequence. Our data showed that TSG101 knockdown decreased the extracellular HBV DNA level and led to accumulation of capsids in cells, which supported that TSG101 was required for HBV capsid extracellular export. Complementation of TSG101 with its mutant form harboring ubiquitin-binding deficiency cannot rescue HBV production further proved that TSG101 with the

function of sorting ubiquitylated substrates was crucial for HBV egress. Direct evidence for TSG101 mediating HBV egress is the observation that TSG101 knockdown led to the accumulation of virions or virion-like particles in MVBs. The HBc without Lysine residues, particularly the Lys-96 residue, abrogated the interaction of HBc and TSG101. Of note, the K96R mutation in HBc did not affect the capsids assembly but remarkably reduced the HBV production, suggesting that Lys-96 is essential for TSG101 recognition thereby affects HBV egress.

TSG101 was originally defined as a tumor susceptibility gene and later characterized to be a core component of the ESCRT-I complex, and functions in the vacuolar protein sorting pathway involved in trafficking targets for lysosomal degradation [27,28]. Accumulating evidences revealed that several viruses hijack TSG101 and the vacuolar protein sorting pathway for assembly and budding. It has been reported that human immunodeficiency virus type 1 structural precursor polyprotein Gag interacts with TSG101 to link ESCRT machinery to viral budding and egress [29]. Similarly, Rous sarcoma virus, equine infectious anemia virus and murine leukemia virus, which also belong to Retroviridae, depend on TSG101 to bud and egress [30]. In addition, TSG101 can facilitate influenza A virus trafficking, but this is effectively restricted by the interferon-stimulated gene 15 [31]. A recent paper showed that TSG101 contributed to virion formation of porcine reproductive and respiratory syndrome virus via interaction with the nucleocapsid protein [32]. Although a previous study demonstrates that TSG101 does not affect HBV production in plasmid transfection system [33], different observations were obtained in HBV replicating and infection cells. Here, we report that the initiation of HBV egress involves TSG101. Together, these studies suggested that viruses from different families have evolved to use a general mechanism to egress.

NEDD4 has been described to be involved in HBV replication [10,34]. Consistently, our data showed that knockdown of NEDD4 or complementing a mutant without enzyme activity limited HBV production. We further dissected the molecular mechanism. NEDD4 knockdown did not affect the intracellular HBV RNA and HBc levels. More importantly, we find that NEDD4 knockdown mitigated HBc ubiquitination and HBc-TSG101 interaction, which resulted in reduced HBV particles count in MVBs. We also provide direct evidence that the PPAY motif of HBc is required for NEDD4 recruitment which is essential for subsequent TSG101 recognition. Either the deficiency of the endosomal E3 ubiquitin ligase recruitment or the mutation of key ubiquitination site in HBc without affecting capsid assembly leads to a sharp decrease of HBV production. These data implicate an important role for NEDD4 in mediating the interaction between HBc-TSG101 thereby the HBV release.

As a pleiotropic structure protein, HBc participates in many important steps of HBV lifecycle [35]. HBc interacts with importin-β and directs the transport of the capsid to nuclear pore and mediates the rcDNA nuclear release for cccDNA synthesis [36]. HBc also actively participates in viral second-strand DNA synthesis via interaction with Pol pgRNA/DNA along with essential host factors [37,38]. It has been reported that HBc directs the envelopment of rcDNA containing capsids assembly and secretion of virion. However, its roles in HBV particle egress have not been fully illustrated. Our study broadens the current understanding of HBc functions in HBV lifecycle and highlights the role of HBc in the trafficking of mature HBV particle.

In our study, not only HBV virion, but also HBsAg which mainly represent subviral particles were regulated by TSG101. It is consistent with recent studies, which demonstrate that MVB is involved in HBsAg trafficking [39,40]. TSG101 is a component of ESCRT regulating MVB biogenesis [41]. TSG101 is also a cargo sorting substrate into exosome and is commonly used as exosome marker [19]. Exosome are derived within MVBs [42]. Additionally, a fraction of HBV virions and subviral particles can be released as exosomes [12,43]. These evidences imply that TSG101 may influence exosome mediated HBV virion and subviral particle release. Further investigation is needed.

## Materials and methods

### Ethics statement

Animals used in this study were treated in accordance with the guidelines on humane care, and the protocols were approved by the Ethic Committee of Animal Facility, Wuhan University.

### Cell lines

Huh7 cells, HepAD38 cells, HepG2-NTCP cells, HEK293T cells, and HEK293A cells were cultured in Dulbecco's modified Eagle's medium (DMEM, Bio-channel, Nanjing, China) supplemented with 10% fetal bovine serum (FBS, Lonsera, Canelones, Uruguay), 100 U/ml penicillin, and 100 μg/ml streptomycin sulfate (S1 Table, Gibco, Grand Island, New York, USA) at 37°C in a 5% $CO_2$ incubator. Mycoplasma contamination surveillance was performed routinely.

### HBV transgenic mice

C57B/6N-HBV1.28 HBV transgenic mice aged 6–8 weeks were purchased from Beijing Vitalstar Biotechnology Cooperation Limited. Blood was taken from the orbit and the serum levels of HBsAg and HBV DNA were determined. Eight mice were randomly divided into two groups. The recombinant AAV expressing control shRNA or Tsg101-targeted shRNA (S2 Table, $1 \times 10^{12}$ viral genome, diluted in 200 μl PBS (Sigma-Aldrich, United Kingdom)) were delivered into peripheral blood by tail intravenous injection. Serum levels of HBsAg and HBV DNA were determined at 7 days and 14 days after the first administration. Levels of Tsg101 in liver were determined by WB at 14 days after the first administration.

### Immunoprecipitation-mass spectrometry

The cells were harvest at 72 hours after transfection and were suspended in the weak right inferior phrenic arteries (RIPA) lysis buffer. The cell lysates were pre-cleared with protein A/G agarose beads immunoprecipitated with an anti-Myc monoclonal antibody or control IgG (S1 Table) coated agarose beads (60 μl) at 4°C for 12 h. Following 3 washes, the bound proteins were eluted with 100 μl 0.2M glycine (pH = 2) and neutralized with 40 μl Tris (pH = 8). The captured proteins were digested in urea buffer as previously described [44]. The obtained peptides were desalted with the C18 stage tips (Thermo Fisher Scientific) and analyzed by an Orbitrap Exploris 480 mass spectrometer with the FAIMS Pro interface. The samples were analyzed as previously described [45]. Raw data are available on ProteomeXchange Consortium (http://www.proteomexchange.org, submission reference number: 1-20220513-61674).

### CsCl density gradient centrifugation

The culture supernatants of HepAD38 (250 ml) were filtered with 0.22 μm filter and concentrated with Amicon Ultra-15 centrifugal filter unit (Millipore, Sigma, Burlington, MA, USA). Final volumes were normalized to 2.8 ml. Discontinuous CsCl density gradients (1.1, 1.2, 1.3, 1.4, and 1.5 g/cm$^3$) were prepared with a solution containing 20 mM Tris-HCl (pH 7.5) as previously reported [46]. The concentrations (1.5 ml) were laid on the CsCl gradient, and centrifuged at 22,000 rpm for 20 h at 4°C in a Beckman rotor (Beckman Coulter, Fullerton, CA, USA).

### Co-immunoprecipitation

Cells were lysed with weak RIPA lysis buffer containing protease inhibitors cocktail. Particularly, the cell lysates used for the interaction of proteins with NEDD4 were prepared with the

IP lysis buffer (25 mM Tris-HCl pH = 7.4, 150 mM NaCl, 1 mM EDTA, 1% NP-40, and 5% glycerol) supplemented with protease inhibitors cocktail. The lysates centrifuged (12, 000 rpm) for 10 min at 4°C after a 10 min ice incubation. The cell lysates were pre-cleared with protein A/G agarose beads and immunoprecipitated with target gene protein antibody or control IgG (S1 Table) coated agarose beads (60 μl) at 4°C for 12 h. Following 3 washes, the beads were heated in SDS loading buffer at 100°C for 10 min and supernatants were collected for immunoblotting assay.

## Transmission electron microscopy

The cells were trypsinized and washed 3 times with precooled phosphate-buffered saline (PBS) and were collected with centrifugation. The pellets were fixed with 2.5% glutaraldehyde in PBS at room temperature for 2 hours. The fixed pellets were rinsed with PBS, 3 times for 10 min each and were treated with 1% osmium tetroxide in PBS for 1 h. After 3 times rinsing, the pellets were dehydrated through a graded series of ethanol concentrations (50, 70, 90 and 100%) for 10 min each prior to embedment with resin. The samples were sliced into 50 nm sections and analyzed with transmission electron microscope (HITATI 7800).

## Focused ion beam-scanning electron microscopy (FIB-SEM)

Cells were washed with PBS twice and immediately fixed in 2.5% glutaraldehyde in 0.1 M phosphate buffer (PB) for 2 hours at room temperature, then incubated for 1 h in 2% (wt/vol) osmium tetroxide and 1.5% (wt/vol) K4[Fe(CN)6] in PB followed by the treatment of 1% (w/v) tannic acid in 100 mM cacodylate buffer for 1 hour, then 30 min in 2% (w/v) osmium tetroxide in water followed by 1% (w/v) uranyl acetate treatment for 2 hours at room temperature. After the dehydration cycles, samples were embedded in Epon-Araldite mix.

Tomography was done with Zeiss Crossbeam 550. Fibbing conditions were: 30 keV, 700 pA, 20 nm slice thickness at a tilt angle of 54 degree and a working distance of 5.0 mm. Images acquired at 10 nm/pixel. The imaging conditions were: 1.5 keV, 1 nA, 8 μs/4.5 μs with a frame size 16 μm × 36.5 μm × 15.3 μm.

## Agarose gel electrophoresis-immunoblot for HBV capsids

Levels of HBV capsids were determined as previously reported [47]. The cells were collected and washed with pre-cooled PBS. The cells were lysed by the buffer (10 mM Tris (pH 7.5), 1 mM EDTA, 50 mM NaCl, 8% sucrose, 0.25% NP-40) supplemented with protease inhibitor cocktail. MgCl$_2$ (at the final concentration 6mM), DNase I (at the final concentration 20 μg/ml), RNase A (at the final concentration 1 mg/ml) were added in the lysates. Supernatants were collected by centrifuge (13, 000 rpm 4°C) after a warm incubation (37°C, 20 min). The lysate was electrophoresed in a 1.5% agarose gel (40 V, 2.5 hours) and transferred to a PVDF membrane (Millipore, Darmstadt, Germany). Then, membranes were blocked with 5% skim milk (Difco, Sparks, Maryland, USA) for 1 hour at room temperature. Next, membranes were incubated with an anti-HBc antibody (S1 Table) overnight at 4°C followed by incubation with secondary antibodies for 1 hour at room temperature. Bands were detected using GeneGnome XRQ chemiluminescence imaging system (GeneGnome, Hong Kong, China).

## Western blotting

Cells and mouse liver tissues were lysed by weak RIPA buffer supplemented with protease inhibitor cocktail. The lysate was electrophoresed in a 12% SDS-PAGE gel and transferred to a polyvinylidene difluoride (PVDF) membrane (Millipore, Darmstadt, Germany). Then,

membranes were blocked with 5% skim milk (Difco, Sparks, Maryland, USA) for 1 hour at room temperature. Next, membranes were incubated with primary antibodies (S1 Table) overnight at 4°C followed by incubation with secondary antibodies for 1 hour at room temperature. Protein bands were detected using GeneGnome XRQ chemiluminescence imaging system (XRQ-NPC, GeneGnome, Hong Kong, China).

## Immunofluorescence

Cells were grown approximately 80% confluent and rinsed with PBS. Neutral formalin solution was applied to fix the cell at room temperature for 20 min. After rinsed 3 times with PBS, the cells were permeabilized with 0.1% Triton X-100 in PBS for 2 min, rinsed with D-PBS, and incubated in blocking buffer (5% normal goat serum in PBS) at 37°C for 1 hour. Cells were incubated in primary antibodies (S1 Table) at 4°C overnight and rinsed repeatedly with PBS before incubating in the appropriate fluorescein-labeled secondary antibody for 1 hour at 37°C. Cells were then washed extensively with PBS. Nuclei were labeled by Hoechst. Samples were observed by confocal microscope (Cat #: LCS-SP8-STED, Leica, Germany).

## Cell transfection

For DNA transfection, PEI MAX 40K (Polysciences, Warrington, Pennsylvania, USA) was used according to the manufacturer's instructions. Briefly, plasmid DNA was diluted with Opti-MEM mixed with PEI (DNA:PEI = 1:3, w/w) and incubated at room temperature for 20 min. Cells were rinsed twice with PBS. The DNA and PEI mixture was added into the transfection medium (DMEM containing 10% FBS without antibiotic). Transfection medium were replaced by complete DMEM medium after overnight incubation. The transfected cell could be treated as needed.

We performed siRNA transfection with Lipofectamine RNAiMAX (Invitrogen, Carisbad, California, USA) according to the manufacturer's instruction. Briefly, cells were plated approximately 75% at the time of transfection. The Lipofectamine RNAiMAX Reagent and siRNA were diluted in Opti-MEM Medium as the manufacturer recommended. The mixture was incubated in room temperature for 15 min and added into the transfection medium (DMEM containing 10% FBS without antibiotic). Transfection medium were replaced by complete DMEM medium after 6~8 hours incubation. The transfected cell could be treated as needed.

## Stable knockdown cell lines preparation

Lentivirus particles expressing TSG101, NEDD4, or NEDD4L targeted shRNA were produced as describe in the section *Virus production and infection*. HepAD38 cells or HepG2-NTCP cells were seeded approximately 70% confluent. Lentivirus infected the cells with culture media containing 8 μg/mL polybrene to improve infection efficiency for 24 hours and the target cells were selected with 2 μg/ml puromycin. Culture medium with 2 μg/ml puromycin was refreshed every other day. The knockdown efficiency of target gene was confirmed basing on the WB assay or qPCR assay.

## Virus production and infection

For AAV production, HEK293T cells were seeded into 15 cm dishes 12 hours before transfection. Plasmids pX552-shRNA (shCtrl or shTsg101), pAAV2/8RC, and pHelper in 1:1:1 molar ratio) were transfected into the cells with PEI MAX 40K when cells grow approximately to 70–80% confluent. The culture supernatants and cell lysates obtained by freeze-thaw method were filtered with 0.22 μm filter 72 h after transfection and concentrated with Vivaspin protein

concentration unit (Cat #: 28932363, Cytiva, USA). The titer of recombinant AAV was determined by qPCR.

For lentivirus production, HEK293T cells were seeded into T25 culture flask (NEST Biotechnology) 12 hours before transfection. 1 μg recombinant pLKO.1-shRNA plasmid or pWPI-target gene overexpressing plasmid was co-transfected with 750 ng psPAX2 packaging plasmid and 250 ng pMD2.G envelope plasmid into the cells with PEI MAX 40K when cells grow approximately to 70–80% confluent. The culture supernatants were collected at 48 hpi. and 72 hpi. and filtered with 0.45 μm filter. The titer of recombinant lentivirus was determined by qPCR.

For lentivirus infection, the target cells were plated 12 hours before infection. Culture medium was refreshed with infection medium (DMEM supplemented with 10% FBS without antibiotics containing 8 μg/ml polybrene (Solarbio, Beijing, China)). Proper dose of lentivirus was added and the infection culture medium was replaced by complete culture medium at 24 hpi. The infected cells were treated as needed.

For HBV production, HBV stocks were obtained from the culture supernatants of HepAD38 cells and was concentrated by a centrifugal filter (Millipore, Ireland). The titer of HBV was determined by an HBV DNA diagnostic kit (S3 Table).

For HBV infection, HepG2-NTCP cells were plated approximately with 90–100% confluent in the culture medium containing 2.5% DMSO for two days before HBV infection. Then the cells were infected with 100 HBV genomes per cell in culture medium containing 5% polyethylene glycol (PEG)-8000 and 2.5% DMSO for 24 hours. The cells were washed 5 times with PBS and cultured in DMEM complete culture medium supplemented with 2.5% DMSO and treated further as indicated in each experiment.

## Enzyme linked immunosorbant assays (ELISA)

HBsAg ELISA (S3 Table) was performed according to the manufacturer's instructions. Levels of HBsAg in HBV transgenic mice serum were determined after 1:4000 dilution.

## Protein expression and purification

6×His tagged NEDD4 was encoded in pET28a and expressed in BL21(DE3) chemically competent cell (Tsingke, Beijing, China). The cells were grown in LB medium containing 50 mg/L kanamycin and the protein expression was induced by 1 mM IPTG at 16˚C for 16 h. Protein purified by the kit BeyoGold His-tag Purification Resin (S3 Table) as the manual provided by the manufacturer.

GST tagged TSG101 UEV domain (residues 2–145) and GST tagged HBc were encoded in pGEX6p-1 respectively and were expressed in Rosetta-gami 2(DE3) chemically competent cells. The cells were cultured in LB medium containing 100 mg/L ampicillin and 34 mg/L chloramphenicol. The expression of the recombined proteins was induced by 1 mM IPTG at 16˚C for 16 h. The proteins were purified by the kit BeyoGold GST-tag Purification Resin (S3 Table) following the protocol provided by the manufacturer. The N-terminal GST tag of the recombined protein was removed by using the kit PreScission Protease (S3 Table) for other experiments.

## Ubiquitination assays

For *in vivo* ubiquitination experiments, cells were lysed in IP lysis buffer and the lysates were immunoprecipitated with anti-HBc antibody-coated agarose beads. The beads eluates were subjected to WB assay. For HBc residues ubiquitination assay and PPAY motif function in HBc ubiquitination assay, Huh7 cells were co-transfected HA-ubiquitin following wild type

HBc or mutant HBc transfection. The immunoprecipitation and WB assay were the same as above.

For *in vitro* ubiquitination experiments, recombind HBc and NEDD4 were expressed by *E. coli* and purified as described in the section *protein expression and purification*. Ubiquitination was analyzed with a ubiquitination kit (S3 Table) following the protocols provided by the manufacturer. The purified NEDD4 was as E3 ubiquitin ligase doner.

## Cell viability assay

Cell viability was determined using Cell Counting Kit-8 (CCK-8) (S3 Table) according to manufacturer's protocol. Briefly, cells were treated as indicated, cell culture medium 1/10th (v/v) CCK-8 solution was added and incubated for 1 hour at 37˚C. The absorbance was measured at 450 nm.

## Quantitative real-time polymerase chain reaction

For relative level of intracellular mRNA assays, total RNA was extracted from cell or mouse liver homogenate by Trizol lysis method. First-strand cDNA synthesis by reverse transcription was performed by using a qPCR RT Master Mix with gDNA Remover kit (S3 Table) according to the manufacture's instruction. Then expression levels of target genes were analyzed with qPCR assays (LightCycler 480 II, Roche, Rotkreuz, Swiss). The sequences of forward primers and reverse primers from 5′ to 3′ were listed in S2 Table. respectively. The mRNA data were calibrated to β-actin via the method of ΔΔCp.

Absolute quantifications of HBV DNA in culture supernatant or mouse serum were performed by using an HBV DNA Diagnostic Kit following manufacture's instruction.

## Quantification and statistical analysis

At least 3 or 4 biological replicates were set for each test for all experiments. Data were presented as mean ± SD or mean ± SEM as indicated. Comparison between two groups was performed by one-way ANOVA. Statistical analysis was conducted with GraphPad Prism 8.0. $^*p < 0.05$; $^{**}p < 0.01$; $^{***}p < 0.001$.

## Supporting information

**S1 Fig. TSG101 is a potential factor regulating HBV replication.** (A) Myc-HBc or HBc expressing plasmid was transfected into Huh7 cells. Intracellular capsids were detected by agarose gel electrophoresis-immunoblot. HepAD38 cells were set as positive control. (B) Myc-HBc was transfected into Huh7 cells. 72 hours after transfection, the cells were suspended with RIPA (weak) lysis buffer. Myc IP experiment was carried out on the cell lysates. IgG IP was set as negative control. Mass spectrometry was performed and identified 92 HBc binding proteins. (C and D) Gene ontology analysis of biological process and cellular component with DAVID bioinformation database for the HBc binding factors. (C) The enrichment of biological processes ranked in top 10 were shown. (D) The enrichment of cellular components ranked in top 10 were shown. (E) HepAD38 cells were transfected with CHP1, EHD1, KHDRBS1, LMAN1, PTPN23, or TSG101 targeted siRNA respectively and maintained in DMEM containing 2% DMSO for 2 days. Levels of the genes mRNA were determined by qPCR assay (% of siRNA). Values show the mean ± SD. $^{**}p < 0.01$, $^{***}p < 0.001$. (TIFF)

**S2 Fig. TSG101 regulated HBV egress and viral antigens release.** (A) HBc IP and TSG101 IP experiments were conducted on HepAD38 cell lysates. IgG IP was set as a negative control.

(B-E) HepAD38 cells with or without stable knockdown of TSG101 were maintained with DMEM supplemented with 2% DMSO for 2 days. (F-I) HepG2-NTCP cells with or without stable knockdown of TSG101 were pretreated with 2.5% DMSO for 2 days following HBV infection at an MOI of 200 and maintained with 2.5% DMSO for 7 days. (B, E, F, and I) Levels of intracellular TSG101 and HBc were determined by WB. (C and G) Levels of HBV DNA and HBsAg in cell culture supernatant were determined by qPCR and ELISA respectively (% of shCtrl). (D and H) Levels of intracellular HBV total RNA and pgRNA were determined by qPCR (% of shCtrl). Values show the mean ± SD. *$p < 0.05$, **$p < 0.01$, ***$p < 0.001$. (J) HepAD38 cells with or without stable knockdown of TSG101 were maintained with DMEM supplemented with 2% DMSO with or without the treatment of 20nM MG132 for 16 hours or 100 nM Bafilomycin A1 for 16 hours. Levels of intracellular L-HBs and HBc were determined by WB.
(TIFF)

**S3 Fig. TSG[+] MVB is required for HBV egress.** (A) HepAD38 cells were treated with or without doxycycline 1 μg/ml for 10 days. The subcellular distribution of TSG101 and CD63 were detected by immunofluorescence (IF) assay. The fluorescence intensity of TSG101 and CD63 along the indicated line were scanned by software ImageJ. Quantification of TSG101[+]CD63[+] puncta. Four random cells with TSG101[+]CD63[+] puncta were observed. Values show the mean ± SEM, *$p < 0.05$. (B-E) HepG2-NTCP cells were pretreated with 2.5% DMSO for 2 days following HBV infection at an MOI of 200 and maintained with 2.5% DMSO. The cell was treated with 1 μg/ml or 5 μg/ml U18666A from 3 days post infection. The cell culture supernatants and the cells were harvested at 7 days post infection. (B) Levels of HBV DNA and HBsAg in cell culture supernatant were determined by qPCR and ELISA respectively (% of NC). (C) Levels of intracellular HBV total RNA and pgRNA were determined by qPCR (% of NC). (D) Levels of intracellular HBc were determined by WB. The intracellular HBV capsids were detected by agarose gel electrophoresis-immunoblot assay. (E) Cell viability was evaluated by CCK-8 assay (% of NC). Values show the mean ± SD. *$p < 0.05$, **$p < 0.01$.
(TIFF)

**S4 Fig. Lysine sites and PPxY motif in HBc.** Sequence analysis for lysine sites (A) and PPxY motif (B) of HBc from different genotype of HBV.
(TIFF)

**S5 Fig. NEDD4 mediates the ubiquitination of HBc and is critical for HBV egress.** (A-C) HepAD38 cells with or without stable knockdown of NEDD4 were maintained with DMEM supplemented with 2% DMSO for 2 days. (D-F) HepG2-NTCP cells with or without stable knockdown of NEDD4 were pretreated with 2.5% DMSO for 2 days following HBV infection at an MOI of 200 and maintained with 2.5% DMSO for 7 days. (G-I) HepAD38 cells with or without stable knockdown of NEDD4L were maintained with DMEM supplemented with 2% DMSO for 2 days. (J-L) HepG2-NTCP cells infected with lentivirus expressing NEDD4L-targeted shRNAs were pretreated with 2.5% DMSO for 2 days following HBV infection at an MOI of 200 and maintained with 2.5% DMSO for 7 days. (A and D) Levels of NEDD4 and HBc were determined by WB. (B, E, H, and K) Levels of HBV DNA and HBsAg in cell culture supernatant were determined by qPCR and ELISA respectively (% of shCtrl). (C, F, I, and L) Levels of intracellular HBV total RNA and pgRNA were determined by qPCR (% of shCtrl). (G and J) Knockdown efficiency of NEDD4L was confirmed by qPCR (% of shCtrl). Values show the mean ± SD. *$p < 0.05$, **$p < 0.01$, ***$p < 0.001$. (M) HBc IP and NEDD4 IP experiments were conducted on HepAD38 cell lysates. IgG IP was set as negative control.
(TIFF)

**S1 Table. Antibodies and chemicals.**
(PDF)

**S2 Table. Oligonucleotides.**
(PDF)

**S3 Table. Commercial Kits.**
(PDF)

**S1 Data. Raw data 1.**
(XLSX)

**S2 Data. Raw data 2.**
(PDF)

## Acknowledgments

We thank the staffs at the Research Center for Medicine and Structural Biology of Wuhan University for technical assistance.

## Author Contributions

**Conceptualization:** Yingcheng Zheng, Yuchen Xia.

**Funding acquisition:** Yuchen Xia.

**Investigation:** Yingcheng Zheng, Mengfei Wang, Sitong Li, Yanan Bu, Zaichao Xu, Guoguo Zhu, Chuanjian Wu, Kaitao Zhao, Aixin Li, Quan Chen, Jingjing Wang, Rong Hua, Yan Teng, Li Zhao.

**Methodology:** Yanan Bu, Zaichao Xu, Guoguo Zhu, Chuanjian Wu, Kaitao Zhao, Aixin Li, Quan Chen, Jingjing Wang, Rong Hua, Yan Teng, Li Zhao.

**Project administration:** Xiaoming Cheng, Yuchen Xia.

**Resources:** Yuchen Xia.

**Supervision:** Yuchen Xia.

**Validation:** Yingcheng Zheng, Mengfei Wang, Sitong Li, Yanan Bu.

**Writing – original draft:** Yingcheng Zheng.

**Writing – review & editing:** Xiaoming Cheng, Yuchen Xia.

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
