## [Decision Letter · Decision Letter 0]

27 Feb 2023

Dear Dr. Xia,

Thank you very much for submitting your manuscript "Hepatitis B Virus hijacks TSG101 to Facilitate Egress Via Multiple Vesicle Bodies" for consideration at PLOS Pathogens. As with all papers reviewed by the journal, your manuscript was reviewed by members of the editorial board and by several independent reviewers. In light of the reviews (below this email), we would like to invite the resubmission of a significantly-revised version that takes into account the reviewers' comments.

We cannot make any decision about publication until we have seen the revised manuscript and your response to the reviewers' comments. Your revised manuscript is also likely to be sent to reviewers for further evaluation.

Sincerely,

Craig Meyers, Ph.D.

Academic Editor

PLOS Pathogens

Alison McBride

Section Editor

PLOS Pathogens

Kasturi Haldar

Editor-in-Chief

PLOS Pathogens

orcid.org/0000-0001-5065-158X

Michael Malim

Editor-in-Chief

PLOS Pathogens

orcid.org/0000-0002-7699-2064

Reviewer's Responses to Questions

**Part I - Summary**

Reviewer #1: Yingcheng Zhang , et al. report in this manuscript the essential role of TSG101 interaction with core protein in HBV virion morphogenesis. Specifically, in agreement with the prior work demonstrating the essential role of Cp K96 (but not K7) in HBV virion production (PMID: 35867543) and later domain activity of Cp 129PPAY132 motif (PMID: 24009707) as well as the critical role of ESCRT complexes in HBV virion budding, this report presents evidence showing that HBV virion production depends on TSG101 interaction with Cp in CpK96-, NEDD4- and later domain motif (Cp 129PPAY132 )-dependent manner. Importantly, the work demonstrated , for the first time, that Cp is ubiquitylated. Overall, the study is well conceived. The results support the notion that HBV virion production depends on the later domain-mediated Cp ubiquitylation by NEDD4 and TSG101 interaction with ubiquitylated Cp. I have the following specific points to help further improving the manuscript.

Reviewer #2: This study assessed the mechanism on how HBV virions were sorted into MVBs for egress, and indicated that TSG101 recognition for NEDD4 ubiquitylated HBc is critical for MVBs-mediated HBV egress.The manuscript is well-organized and clearly stated. Some concerns need to be addressed.

**Part II – Major Issues: Key Experiments Required for Acceptance**

Reviewer #1: 1. It is well known that naked capsids can be secreted (released) into culture media and interfere the accurate quantification of secreted virions by qPCR assays. Because the method for quantification of HBV DNA (virions) in culture media is not provided, it is not clear how this technical issue was addressed in this study.

2. For the results presented in Fig. 4B-D and Fig. 4F-G, intracellular HBV core DNA should be detected to rule out the possibility that the reduced virion production is due to the deficiency of mutant Cp to support pgRNA packaging and/or viral DNA synthesis.

3. For results presented in Fig. 6A, it should be tested whether NEDD4 interaction with HBc is 129PPAY132 motif-dependent.

4. It is not clear if the polyubiquitylation of HBc depends on the 129PPAY132 motif. You can also use your K7R and K96R mutant HBc to determine which lysine residue is ubiquitylated.

5. It is also interesting to know if the HBc ubiquitylation occurs in the context of nucleocapsids, empty capsids and/or free dimers. In another word, is the morphogenesis of complete virions and incomplete virions via the same or distinct mechanism?

Reviewer #2: 1. In most figures (for example Fig 1B), the knockdown efficiency of siRNA should be determined by protein level using Western blot.

2. Line 184: TSG101 ubiquitination at these two sites should be performed.

3. Fig 2B and 2D: Is there any basis for the morphological identification of MVBs in transmission electron microscopy experiments.

**Part III – Minor Issues: Editorial and Data Presentation Modifications**

Reviewer #1: 1. Line 37, the “ESCRT” should be spelled out.

2. The sentence starting at line 70 “However, it is still unclear which host factor(s) recognizes HBV virions and how is the sorting into MVBs orchestrated for HBV to egress the cell.” is not a clear statement. My understanding is that MVBs are the place for HBV virion morphogenesis and sorting/secretion. Accordingly, a better statement is: “it is still unclear which and how host factors recognize HBV capsids and facilitate the morphogenesis and egress of HBV virions from hepatocytes”.

3. Although the result presented in Fig. S1A showed that flagged-tagged HBc assembles into capsids. Do you know the Cp pulled down in your IP-mass spectrometry analysis is in the form of capsids or not?

4. The rationale for the selection of the six proteins for further investigation is not clearly described.

5. Fig. 1H, it appears that knockdown of TSG101 also reduced the production of SVPs. The role of TSG101 in SVP secretion should be more thoroughly discussed.

6. In Fig S3A, in DOX (-) and DOX (+), at least core or other viral markers should be examined to confirm the experimental conditions.

7. In the legend of Fig. 7, is there published evidence for tenatoprazole inhibition of HBV virion production? If yes, it should be cited and discussed in the text.

8. In the sentence from line 285 - 287, the “HBV particles” indicate HBV virions or capsids? The amounts of virions or virion-like particles are reduced in TSG101 knockdown cells (Fig. 2E).

9. Sentence from line 333 - 334, no direct evidence to support this claim.

Reviewer #2: 1. In most figures (for example Fig 1B), the knockdown efficiency of siRNA should be determined by protein level using Western blot.

2. Line 184: TSG101 ubiquitination at these two sites should be performed.

3. Fig 2B and 2D: Is there any basis for the morphological identification of MVBs in transmission electron microscopy experiments.

PLOS authors have the option to publish the peer review history of their article (what does this mean?). If published, this will include your full peer review and any attached files.

Reviewer #1: No

Reviewer #2: No
---

## [Decision Letter · Decision Letter 1]

24 Apr 2023

Dear Dr. Xia,

We are pleased to inform you that your manuscript 'Hepatitis B Virus hijacks TSG101 to Facilitate Egress Via Multiple Vesicle Bodies' has been provisionally accepted for publication in PLOS Pathogens.

Best regards,

Craig Meyers, Ph.D.

Academic Editor

PLOS Pathogens

Alison McBride

Section Editor

PLOS Pathogens

Kasturi Haldar

Editor-in-Chief

PLOS Pathogens

orcid.org/0000-0001-5065-158X

Michael Malim

Editor-in-Chief

PLOS Pathogens

orcid.org/0000-0002-7699-2064

Reviewer Comments (if any, and for reference):

Reviewer's Responses to Questions

**Part I - Summary**

Reviewer #1: The authors addressed my comments with satisfaction. I have no further comment to the revised manuscript.

Reviewer #2: (No Response)

**Part II – Major Issues: Key Experiments Required for Acceptance**

Reviewer #1: No.

Reviewer #2: (No Response)

**Part III – Minor Issues: Editorial and Data Presentation Modifications**

Reviewer #1: No.

Reviewer #2: (No Response)

PLOS authors have the option to publish the peer review history of their article (what does this mean?). If published, this will include your full peer review and any attached files.

Reviewer #1: No

Reviewer #2: No

---

## [Editor Report · Acceptance letter]

9 May 2023

Dear Dr. Xia,

We are delighted to inform you that your manuscript, " Hepatitis B Virus hijacks TSG101 to Facilitate Egress Via Multiple Vesicle Bodies ," has been formally accepted for publication in PLOS Pathogens.

Best regards,

Kasturi Haldar

Editor-in-Chief

PLOS Pathogens

orcid.org/0000-0001-5065-158X

Michael Malim

Editor-in-Chief

PLOS Pathogens

orcid.org/0000-0002-7699-2064